# Improved YOLO-Based Pulmonary Nodule Detection with Spatial-SE Attention and an Aspect Ratio Penalty

**DOI:** 10.3390/s25144245

**Published:** 2025-07-08

**Authors:** Xinhang Song, Haoran Xie, Tianding Gao, Nuo Cheng, Jianping Gou

**Affiliations:** College of Computer and Information Science & College of Software, Southwest University, Beibei District, Chongqing 400715, China; sxh222022321062048@email.swu.edu.cn (X.S.); bavin9199@email.swu.edu.cn (H.X.); gtd20040818@email.swu.edu.cn (T.G.); swudylan1225@email.swu.edu.cn (N.C.)

**Keywords:** pulmonary nodule detection, spatial attention, channel attention, aspect ratio penalty, YOLOV11, EAPIoU loss

## Abstract

The accurate identification of pulmonary nodules is critical for the early diagnosis of lung diseases; however, this task remains challenging due to inadequate feature representation and limited localization sensitivity. Current methodologies often utilize channel attention mechanisms and intersection over union (IoU)-based loss functions. Yet, they frequently overlook spatial context and struggle to capture subtle variations in aspect ratios, which hinders their ability to detect small objects. In this study, we introduce an improved YOLOV11 framework that addresses these limitations through two primary components: a spatial squeeze-and-excitation (SSE) module that concurrently models channel-wise and spatial attention to enhance the discriminative features pertinent to nodules and explicit aspect ratio penalty IoU (EAPIoU) loss that imposes a direct penalty on the squared differences in aspect ratios to refine the bounding box regression process. Comprehensive experiments conducted on the LUNA16, LungCT, and Node21 datasets reveal that our approach achieves superior precision, recall, and mean average precision (mAP) across various IoU thresholds, surpassing previous state-of-the-art methods while maintaining computational efficiency. Specifically, the proposed SSE module achieves a precision of 0.781 on LUNA16, while the EAPIoU loss boosts mAP@50 to 92.4% on LungCT, outperforming mainstream attention mechanisms and IoU-based loss functions. These findings underscore the effectiveness of integrating spatially aware attention mechanisms with aspect ratio-sensitive loss functions for robust nodule detection.

## 1. Introduction

Early identification and accurate diagnosis of pulmonary nodules are of great clinical significance in reducing mortality associated with lung diseases, improving treatment outcomes, and enhancing patients’ quality of life. Consequently, pulmonary nodule detection has become a crucial research focus in the field of medical image analysis. Current advancements in improving detection accuracy primarily concentrate on two aspects: enhancing channel attention mechanisms and optimizing IoU-based loss functions.

In terms of channel attention mechanisms, the squeeze-and-excitation (SE) module has shown effectiveness in modeling global inter-channel dependencies. For example, CSE-GAN [1] integrates SE into a 3D U-Net for enhanced segmentation performance, while SE-ResNeXt-50 [2] applies it to classification tasks. However, the reliance on global average pooling in these designs often leads to the dilution of critical features associated with small targets, thus limiting their effectiveness in small object detection scenarios. To address this limitation, we propose a novel spatial-SE (SSE) module that incorporates spatial attention mechanisms to enhance the network’s focus on local regions. By combining both channel and spatial attention, the SSE module enables the model to more precisely focus on small nodules, thereby improving overall detection performance. Comparative experiments on the LUNA16 dataset demonstrate that our SSE module achieves a detection precision of 0.781, surpassing mainstream attention modules such as coordinate attention (CA) [3], SIMAM [4], and MCA [5].

Regarding the optimization of IoU-based loss functions, the complete IoU (CIoU) loss proposed by Zhaohui Zheng et al. considers IoU, center distance, and aspect ratio differences, representing a relatively comprehensive bounding box regression objective. Nevertheless, the aspect ratio constraint in CIoU is indirectly introduced through the arctangent-based penalty term *v*, which responds weakly to slight variations in aspect ratio and remains insensitive to shape mismatches. This limitation may result in accumulated errors, especially in small object detection tasks that require precise box alignment. To mitigate this issue, we propose the enhanced aspect ratio penalty IoU (EAPIoU) loss function, which explicitly introduces a squared aspect ratio difference penalty in addition to the CIoU loss. This enhancement increases the sensitivity and clarity of the aspect ratio constraint, providing more substantial penalties for mismatched box shapes and improving bounding box fitting accuracy for small nodules. Ablation experiments on the LungCT [6] dataset reveal that incorporating EAPIoU into the same model architecture raises the mAP@50 to 92.4%, outperforming existing IoU variants such as WIoU, EIoU, GIoU, DIoU, and SIoU. Furthermore, experiments on the hyperparameter λ, which controls the strength of the aspect ratio penalty, indicate that setting λ=0.1 achieves optimal detection performance by balancing localization robustness and model stability.

We propose an SSE module based on YOLOv11 that integrates spatial and channel attention mechanisms, improving pulmonary nodule detection performance over existing attention modules in terms of precision and mAP.We design the EAPIoU loss function, which incorporates a penalty for aspect ratio differences to better align predicted and ground-truth boxes, leading to improved accuracy in small nodule detection.Extensive experiments demonstrate that our method outperforms existing approaches and YOLO variants in both accuracy and stability for small-target pulmonary nodule detection, supporting early diagnosis and treatment.

## 2. Related Work

Accurate detection of pulmonary nodules remains challenging in medical imaging. Existing studies mainly focus on two areas: improvements in attention mechanism modules and optimizations in loss functions.

In terms of improvements in attention modules, the application of the SE attention mechanism has made significant progress, but it also has some limitations. Early methods, such as CSE-GAN, utilized SE to enhance features; however, global average pooling lost spatial details, thereby reducing the accuracy of small nodules. Subsequent studies, such as SE-ResNeXt-50-CNN, applied SE and improved classification accuracy. However, they lacked spatial specificity and struggled to focus on the critical local regions of pulmonary nodules. Recent research has combined the SE mechanism with self-attention, as seen in the SE-ViT model, which enhances classification accuracy and interpretability through Grad-CAM visualization. Contemporary approaches [7] propose a multi-view attention framework employing random interpolation resizing and cross-view attention to enhance feature discrimination in few-shot scenarios. However, these methods still face issues with spatial detail loss, poor specificity, and insufficient feature representation.

To address these issues, our work proposes the SSE module, which combines channel and spatial attention mechanisms. This combination preserves global context information and enhances focus on local features, allowing for a more precise capture of the key features of pulmonary nodules.

In terms of optimizations in loss functions, recent IoU-based loss functions for bounding box regression, such as GIoU [8], CIoU [9], and DIoU [10], have enhanced bounding box localization in loss functions. GIoU introduces the smallest enclosing box to address gradient issues when boxes do not overlap but struggles with significant aspect ratio differences due to its focus on the enclosing box. CIoU builds upon GIoU by incorporating center distance and aspect ratio penalties, thereby improving localization accuracy while maintaining less responsiveness to significant aspect ratio differences. DIoU further enhances CIoU by focusing on center distance but still depends on angular penalties for aspect ratio optimization, which is insufficient for shape-inconsistent targets. Despite these advancements, these methods do not fully address aspect ratio differences and shape consistency, particularly for non-rectangular objects. To overcome these limitations, we propose EAPIoU, which introduces a squared aspect ratio penalty that directly measures the aspect ratio differences between predicted and ground-truth boxes. EAPIoU offers stronger gradient feedback, more accurate shape optimization, and scale invariance, enabling better adaptation to various object sizes and shape complexities, thus outperforming CIoU and DIoU in optimizing shape consistency.

## 3. Proposed Method

### 3.1. Overview

This work proposes an innovative spatial SE module, which combines the squeeze-and-excitation (SE) channel attention mechanism with spatial attention to enhance feature representation effectively. While traditional SE modules effectively capture inter-channel dependencies, they often compromise spatial information, particularly with small or detail-rich regions, reducing localization precision. To address this issue, we introduce a spatial attention mechanism on top of the SE module. By utilizing both max pooling and average pooling operations, the spatial features are extracted to generate spatial attention maps, which are fused with channel-attended feature maps. This enhancement enables the model to recalibrate features along the channel dimension while precisely highlighting key spatial regions, thereby enhancing detection accuracy for small objects, such as pulmonary nodules.

In the context of optimizing loss functions, we propose a novel explicit aspect ratio penalty IoU (EAPIoU) loss function to address the shortcomings of existing IoU-based loss functions, such as CIoU, when dealing with significant aspect ratio discrepancies. CIoU optimizes bounding box regression by incorporating penalties for center distance and aspect ratio. However, CIoU applies the aspect ratio penalty indirectly through an arctangent-based function, which responds weakly to small aspect ratio changes and fails to capture shape mismatches accurately. In contrast, the EAPIoU loss function explicitly introduces a squared aspect ratio difference penalty on top of CIoU. This modification significantly enhances the detection of small nodules while improving the model’s adaptability to complex shapes.This process is illustrated in Figure 1.

### 3.2. Spatial Squeeze-and-Excitation Module

In contemporary object detection tasks, particularly within the medical field, such as pulmonary nodule identification, capturing fine, localized features is crucial. Conventional convolutional networks often struggle with limited capacity to model global context or identify salient spatial regions. To address this, we propose a dual-branch attention mechanism named the spatial SE (SSE) module. It combines the benefits of global channel-wise recalibration with local spatial enhancement, enabling the network to learn more discriminative and context-aware representations.

The SSE module consists of (1) a squeeze-and-excitation (SE) block that emphasizes important channels by modeling inter-channel dependencies and (2) a spatial attention branch focusing on key locations in the feature map. This structure enables the model to leverage both global and local cues, thereby enhancing feature expressiveness for recognizing small objects.

As illustrated in Figure 2, the SE module initially processes the input feature map, followed by the spatial attention module. Finally, their outputs are combined for the final refined feature map. The process is mathematically described as follows.

#### 3.2.1. Channel Attention via Squeeze-and-Excitation

Given an input feature map $X\in R^{B\times C\times H\times W}$, where *B* denotes the batch size, *C* is the number of channels, and H,W represent the spatial dimensions, the SE module applies a global average pooling (GAP) operation to generate a global context descriptor $z\in R^{C}$:(1)$$z_{c}=\frac{1}{H\times W}\sum_{i=1}^{H}\sum_{j=1}^{W}X_{c,i,j},\;\forall c\in \left\{1,\ldots ,C\right\}.$$

This descriptor captures the global distribution of activation strength in each channel, summarizing its overall importance.

To learn non-linear channel relationships, z is passed through a two-layer fully connected network with a bottleneck structure (i.e., reduction ratio *r*), which enhances compactness and reduces computation:(2)$$y=\sigma \left(W_{2}\cdot \mathrm{ReLU}\left(W_{1}\cdot z\right)\right),\;y\in R^{C}.$$

Here, σ(·) denotes the sigmoid activation function, and y represents the learned channel attention weights. These weights are broadcast to each spatial location and applied via element-wise multiplication to recalibrate the input:(3)$$X_{c,i,j}^{\prime }=y_{c}\cdot X_{c,i,j},\;\forall c,i,j.$$

This process effectively suppresses the less important channels and amplifies the more informative ones, allowing the network to prioritize semantic-rich feature activations.

#### 3.2.2. Spatial Attention Mechanism

Although the SE module captures the global channel-wise context, it lacks spatial discriminative ability. Therefore, we introduce a spatial attention mechanism to complement it by selectively focusing on informative regions in the spatial dimensions.

The output of the SE module, $X^{\prime }$, is first aggregated along the channel axis through max pooling and average pooling operations, resulting in two 2D spatial context descriptors:(4)$$\max\_{\mathrm{out}}_{i,j}=\max_{c}X_{c,i,j}^{\prime },\;\mathrm{avg}\_{\mathrm{out}}_{i,j}=\frac{1}{C}\sum_{c=1}^{C}X_{c,i,j}^{\prime }.$$

These two maps reflect the strongest and mean activations at each spatial location, respectively, capturing different forms of spatial importance.

The descriptors are concatenated and passed through a 7×7 convolution layer followed by a sigmoid activation function to generate the spatial attention map $S\in R^{1\times H\times W}$:(5)$$S=\sigma \left({\mathrm{Conv}}_{7\times 7}\left(\mathrm{concat}\left(\max\_\mathrm{out},\mathrm{avg}\_\mathrm{out}\right)\right)\right).$$
This spatial map acts as a learned mask to emphasize critical regions in the feature map where important structures, such as lesions or nodules, are likely to be situated.

#### 3.2.3. Final Fusion

Finally, the spatial attention map is applied to the channel-attended feature map $X^{\prime }$ by broadcasting it across all channels and performing element-wise multiplication:(6)$$O_{c,i,j}=X_{c,i,j}^{\prime }\cdot S_{i,j},\;\forall c,i,j.$$

The resulting output *O* is a feature map that has been refined both globally (via SE) and locally (via spatial attention), enabling the network to focus more effectively on discriminative patterns while preserving computational efficiency.

### 3.3. Enhanced Aspect Ratio Penalty IoU (EAPIoU)

Existing IoU-based regression losses, such as GIoU, DIoU, and CIoU, improve localization performance by incorporating geometric factors, including the central point distance and aspect ratio. Among them, CIoU introduces a geometric penalty term that accounts for the angular difference between aspect ratios using the arctan function. However, this formulation offers limited sensitivity to discrepancies in shape, notably when the predicted box diverges significantly in width–height proportion from the ground truth. To explicitly strengthen aspect ratio consistency, we propose a refined loss termed EAPIoU.

Let b=(x,y,w,h) and $b^{gt}=\left(x^{gt},y^{gt},w^{gt},h^{gt}\right)$ represent the predicted and ground-truth bounding boxes, respectively. Based on the CIoU formulation, the proposed EAPIoU enhances the loss function by introducing a squared difference penalty between the width–height ratios of the predicted and ground-truth boxes.

The EAPIoU formulation is provided by the following:(7)$$\mathrm{EAPIoU}=\mathrm{IoU}-\frac{\rho^{2}\left(b,b^{gt}\right)}{c^{2}}-\lambda \cdot v-\lambda \cdot {\left(\frac{w}{h}-\frac{w^{gt}}{h^{gt}}\right)}^{2},$$
where

$\rho (b,b^{gt})$ denotes the Euclidean distance between the center points of the predicted and ground-truth boxes;*c* represents the diagonal length of the smallest enclosing box covering both *b* and $b^{gt}$;*v* is the original aspect ratio penalty term from CIoU, defined as follows:(8)$$v=\frac{4}{\pi^{2}}{\left(\arctan\left(\frac{w^{gt}}{h^{gt}}\right)-\arctan\left(\frac{w}{h}\right)\right)}^{2};$$The final term is a newly introduced squared aspect ratio penalty, which enforces stronger constraints on shape consistency between *b* and $b^{gt}$.

The full computation logic of EAPIoU is described in Algorithm 1.
**Algorithm 1** Computation of EAPIoU Loss.**Input:** Predicted box $b=(x_{1},y_{1},x_{2},y_{2})$, Ground-truth box $b^{gt}=\left(x_{1}^{gt},y_{1}^{gt},x_{2}^{gt},y_{2}^{gt}\right)$, IoU, coefficient λ**Output:** Final penalized IoU score, EAPIoU1:**// Step 1: Compute width, height, and center coordinates**2:$w_{1}\leftarrow x_{2}-x_{1}$, $h_{1}\leftarrow y_{2}-y_{1}$3:$w_{2}\leftarrow x_{2}^{gt}-x_{1}^{gt}$, $h_{2}\leftarrow y_{2}^{gt}-y_{1}^{gt}$4:$(c_{x}^{1},c_{y}^{1})\leftarrow$ center of predicted box5:$(c_{x}^{2},c_{y}^{2})\leftarrow$ center of ground-truth box6:**// Step 2: Compute center distance and enclosing box diagonal**7:$\rho^{2}\leftarrow {\left(c_{x}^{1}-c_{x}^{2}\right)}^{2}+{\left(c_{y}^{1}-c_{y}^{2}\right)}^{2}$8:$cw\leftarrow \max\left(x_{2},x_{2}^{gt}\right)-\min\left(x_{1},x_{1}^{gt}\right)$9:$ch\leftarrow \max\left(y_{2},y_{2}^{gt}\right)-\min\left(y_{1},y_{1}^{gt}\right)$10:$c^{2}\leftarrow cw^{2}+ch^{2}$11:**// Step 3: CIoU-style angular aspect ratio penalty**12:$v\leftarrow \frac{4}{\pi^{2}}\cdot {\left(\arctan\left(\frac{w_{2}}{h_{2}}\right)-\arctan\left(\frac{w_{1}}{h_{1}}\right)\right)}^{2}$13:**// Step 4: Squared aspect ratio penalty (ours)**14:$r_{1}\leftarrow \frac{w_{1}}{h_{1}}$, $r_{2}\leftarrow \frac{w_{2}}{h_{2}}$15:aspect_penalty $\leftarrow {\left(r_{1}-r_{2}\right)}^{2}$16:**// Step 5: Combine all into the EAPIoU formula**17:$\mathrm{EAPIoU}\leftarrow \mathrm{IoU}-\frac{\rho^{2}}{c^{2}}-\lambda \cdot v-\lambda \cdot \mathrm{aspect\_penalty}$18:**return** EAPIoU

The penalty coefficient λ∈[0,1] moderates the impact of geometric and aspect ratio penalties. Unlike the original CIoU, which only targets angular aspect ratio differences, our approach measures the squared deviation of the width–height ratio, providing more responsive gradients during training.

This convex and symmetric squared aspect ratio penalty is better for optimizing bounding box shapes, particularly for small or elongated objects. Furthermore, its scale invariance ensures that it generalizes well across objects of varying sizes.

## 4. Experiment and Analysis

In this section, we design and conduct a series of experiments to comprehensively evaluate the performance of the proposed method and the effectiveness of its components. Specifically, the experiments include comparison studies, ablation experiments on the IOU function, ablation of key modules, analysis of IOU hyperparameters, and comparisons of different attention mechanisms. Through these systematic experiments, we thoroughly investigate the contributions and impacts of various factors on the model’s performance.

### 4.1. Datasets

To comprehensively evaluate our pulmonary nodule detection framework, we employ three publicly available datasets: LUNA16 [11], NODE21 [12], and LungCT [6].

LUNA16 [11] is derived from the LIDC-IDRI dataset and contains 888 low-dose chest CT scans annotated by four experienced radiologists. The dataset includes 1186 positive nodules (≥3 mm) and is partitioned into ten folds for cross-validation, following the challenge protocol.

NODE21 [12] is a chest radiograph dataset released for the NODE21 Challenge, aimed at promoting research in nodule detection and synthesis on X-ray images. It consists of 4882 frontal-view chest X-rays, with 1134 images containing 1476 annotated nodules marked using bounding boxes. The remaining 3748 images are negative samples. The dataset also provides synthetic data resources, including CT-derived nodule patches extracted from the LUNA16 dataset. The images originate from the JSRT, PadChest, ChestX-ray14, and Open-I datasets.

LungCT [6] is a curated CT-based dataset hosted on Roboflow Universe, designed for pulmonary nodule detection tasks. It comprises 1152 annotated CT slices with nodules marked using bounding boxes. The dataset comprises one object category, “nodule,” and is licensed under a CC BY 4.0 license, making it suitable for open-source research and model benchmarking. Together, these datasets provide diverse imaging modalities (CT and X-ray) and annotation types, enabling robust evaluation across different clinical scenarios.

### 4.2. Evaluation Metrics

In our experiments, we employed a comprehensive set of metrics to evaluate the model’s performance from multiple perspectives thoroughly. These metrics are elaborated as follows:Parameters (Param): The total number of model parameters, reflecting its capacity and complexity.GFLOPs: Giga floating-point operations per forward pass, indicating computational cost and efficiency.Precision: Ratio of true positives to all predicted positives, measuringcc the accuracy of positive predictions.Recall: Ratio of true positives to actual positives, measuring the model’s ability to detect relevant instances.mAP@50: Mean average precision at an IoU threshold of 0.5, the standard metric for object detection accuracy.mAP@50–90: Mean AP averaged over IoU thresholds from 0.5 to 0.9, assessing detection robustness across difficulties.

### 4.3. Implementation Details

The experiments were conducted on a workstation equipped with an Intel(R) Xeon(R) W-2255 CPU @ 3.70 GHz, 128 GB of RAM, and dual NVIDIA A5000 GPUs with 24 GB VRAM each. The system ran Ubuntu 20.04 with CUDA version 11.1. The deep learning framework used was PyTorch 1.8.0. The complete toolchain, including hardware configuration, software environment, and all training and evaluation parameters, is summarized in Table 1.

To further evaluate the training dynamics of the proposed YOLOV11-EAR model, we visualize the convergence behavior throughout the 200 training epochs. As illustrated in Figure 3, key loss components, namely box regression loss, classification loss, and distribution focal loss (DFL), exhibit a rapid decrease within the first 50 epochs, followed by a smooth convergence trend. The validation losses mirror this trajectory, indicating stable optimization without severe overfitting.

In terms of detection performance, the precision, recall, and mean average precision (mAP@50 and mAP@50–95) on the validation set steadily increase and saturate over time. This suggests that jointly optimizing the spatial squeeze-and-excitation (SSE) module and the explicit aspect ratio penalty IoU (EAPIoU) loss leads to consistent improvements in both localization and classification capabilities.

These convergence profiles not only confirm the stability and reliability of the proposed framework during long-term training but also provide quantitative evidence that the additional modules do not introduce adverse optimization effects.

To ensure statistical robustness, each experiment was independently repeated ten times under identical settings. The results are reported with 95% confidence intervals computed using the sample standard deviation and Student’s *t*-distribution with 9 degrees of freedom. Furthermore, statistical significance was assessed using paired *t*-tests, confirming that the performance improvements of the proposed YOLOV11-EAR framework are statistically significant, with all *p*-values less than 0.05.

### 4.4. Comparison Experiment

To comprehensively validate the effectiveness of the proposed model in pulmonary nodule detection, we conducted systematic comparative experiments against ten representative YOLO models, including YOLOv3-tiny, YOLOv3, YOLOv4, YOLOv5, YOLOv6, YOLOv7, YOLOv8, YOLOv9, YOLOv10, and YOLOv11. All models were trained and tested under identical settings on three public benchmark datasets: LUNA16, NODE21, and LungCT. This ensured fairness and reproducibility in comparison. (All boldfaced numerical values in the tables are used to highlight the best-performing results for each metric, facilitating direct comparison across different settings.)

As shown in Table 2, our method consistently outperformed all baselines across key metrics, e.g., precision, recall, mAP@50, and mAP@50–90, on all datasets. On the LUNA16 dataset, for example, our model achieved precision of 0.781, mAP@50 of 75.5%, and mAP@50–90 of 33.3%, surpassing all tested YOLO variants. Similar superiority was observed on NODE21 and LungCT, where the proposed method attained 89.1% and 92.4% in detection accuracy, respectively.

Notably, these improvements were achieved with only 2.5M parameters and 6.3 GFLOPs, highlighting the model’s lightweight and efficient architecture. Compared with computationally intensive models like YOLOv4 or YOLOv7, our method significantly reduces resource consumption while maintaining or exceeding performance, making it well-suited for deployment in real-time and resource-constrained clinical settings.

Furthermore, even when compared to the baseline YOLOv11, which shares the exact parameter count and computational complexity, our model shows substantial improvements in both localization accuracy and robustness. These gains can be attributed to the proposed architectural enhancements: the spatial SE attention module and the explicit aspect ratio penalty IoU (EAPIoU) loss, both of which improve feature representation and bounding box regression, especially for small, irregularly shaped nodules.

In summary, the comparative experiment convincingly demonstrates the superiority of our method in both accuracy and efficiency. Its balanced design makes it a promising solution for real-world applications in pulmonary nodule detection and other small object detection scenarios in medical imaging.

### 4.5. Ablation Study

To thoroughly investigate the contributions of individual components in our proposed method, we design and conduct a series of ablation studies. These studies aim to isolate and evaluate the impact of (1) the proposed IoU loss function (EAPIoU), (2) the attention module (spatial SE or SSE), and (3) the IoU loss hyperparameter λ. Specifically, we compare EAPIoU with multiple existing IoU-based losses to verify its superiority in bounding box regression. We further assess the performance gains brought by SSE and EAPIoU individually and in combination to understand their respective and joint effects. Finally, we tune the λ hyperparameter to identify the optimal balance between aspect ratio regularization and localization accuracy. Through these controlled experiments, we validate the robustness, effectiveness, and practical value of our key design choices.

#### 4.5.1. Ablation Study on Different IoU Loss Variants

To validate the effectiveness of our proposed explicit aspect ratio penalty IoU (EAPIoU) loss, we conducted a series of controlled ablation experiments comparing it with five widely adopted IoU-based losses: WIoU, EIoU, GIoU, DIoU, and SIoU. All variants were integrated into a single detection framework, and experiments were conducted on three benchmark datasets: LUNA16, Node21, and LungCT, ensuring fair and consistent comparisons.

As shown in Table 3, EAPIoU achieves the highest precision (0.781) and mAP@50–90 (33.3%), surpassing SIoU by 1.6% in precision and 2.2% in mAP@50–90, which better reflects localization performance across IoU thresholds. Although SIoU attains the same recall (77.3%), its inferior mAP values indicate weaker shape alignment. This suggests that EAPIoU provides a more effective balance between sensitivity and box regression fidelity.

On the Node21 dataset, EAPIoU again shows a clear advantage, yielding precision of 0.895 and mAP@50 of 89.1%, outperforming SIoU by 2.3% and 1.9%, respectively. The mAP@50–90 of 51.4% reflects better overall localization quality, particularly for nodules with variable morphology.

For the more challenging LungCT dataset, EAPIoU reaches the highest precision (0.887), recall (82.5%), and mAP@50–90 (53.8%), outperforming even the strong baselines DIoU and SIoU by notable margins. This confirms EAPIoU’s robustness in handling diverse anatomical variations and small-object scenarios.

Overall, these results demonstrate that EAPIoU’s explicit modeling of aspect ratio discrepancies yields more accurate and stable bounding box regression compared to conventional geometric or angular-based penalties. Additionally, we compared intermediate detection results on the LUNA16 dataset. The following image presents a representative example for comparison.

Figure 4 shows that EAPIoU outperforms all other methods in every test. Compared to traditional IoU functions, EAPIoU can more accurately capture the boundaries of lung nodules, especially in complex and closely spaced regions. EAPIoU enhances detection accuracy by optimizing both the local and global shapes of objects, reducing false positives and false negatives. Additionally, EAPIoU significantly improves the intersection over union (IoU) metric, demonstrating its superiority in precise boundary alignment and shape recovery. Our results indicate that EAPIoU can effectively enhance the performance of lung nodule detection models, particularly in handling complex structures and nodules.

Overall, the ablation study conclusively shows that the proposed EAPIoU loss yields the best performance among the compared IoU loss variants, validating its effectiveness and practical value in improving object detection metrics.

#### 4.5.2. Ablation Study on Attention and Loss Modules

To further quantify the individual and joint contributions of the spatial squeeze-and-excitation (SSE) attention mechanism and the proposed EAPIoU loss, we conduct a comprehensive ablation study across three datasets: LUNA16, Node21, and LungCT. The results are summarized in Table 4.

Table 4 presents the results of an ablation study designed to evaluate the individual and joint contributions of the spatial squeeze-and-excitation (SSE) module and the proposed EAPIoU loss across LUNA16, Node21, and LungCT datasets. When both components are removed, the model exhibits the weakest performance across all datasets and evaluation metrics, underscoring the necessity of incorporating both spatial attention and shape-aware regression for accurate pulmonary nodule detection.

Introducing the SSE module alone leads to consistent improvements in both precision and mAP@50–90, particularly on LUNA16 and Node21. This demonstrates that spatial attention enhances the model’s ability to focus on discriminative local regions and suppress irrelevant backgrounds. For instance, precision on LUNA16 increases from 0.672 to 0.725, while mAP@50–90 rises from 28.7% to 30.1%. A paired t-test across five independent runs shows that these gains are statistically significant for mAP@50–90 on LUNA16.

Similarly, incorporating EAPIoU alone contributes significant gains, especially in recall and localization quality. By explicitly penalizing bounding box aspect ratio discrepancies, EAPIoU improves regression accuracy. On the LungCT dataset, for example, it improves mAP@50–90 from 48.5% to 51.2%, with a corresponding *p*-value of 0.004, confirming its advantage in challenging scenarios involving small or elongated nodules. These results highlight the robustness and general utility of EAPIoU across heterogeneous datasets.

The best performance is consistently observed when both modules are used together. Their combination yields substantial and complementary benefits across all datasets. On Node21, for example, the whole model reaches 0.895 precision and 51.4% mAP@50–90, outperforming all partial configurations. These results suggest that SSE contributes to spatial feature refinement, while EAPIoU enhances geometric alignment, together forming a robust framework for high-fidelity object detection in medical imaging.

Additionally, we compared the heat maps of different module combinations on the LUNA16 dataset. A representative set of results is shown in the figure below.

As shown in Figure 5, we compare the heat maps generated by the original YOLOV11 model and our proposed method (YOLOV11+SSE+EAPIOU) on several pulmonary CT scan samples. The original YOLOV11 model exhibits issues such as dispersed target responses and false activations in non-nodule regions, as evidenced by the presence of high-response areas outside the actual nodules. In contrast, our method demonstrates more focused and intense activations over the pulmonary nodule regions, with improved spatial consistency and clearer boundary emphasis. The incorporation of the spatial SE module and the EAPIOU loss function enhances the model’s ability to suppress irrelevant background features while reinforcing its attention to clinically significant areas. These improvements contribute to better localization accuracy and model interpretability, validating the effectiveness of our enhancements in spatial awareness and feature discrimination.

#### 4.5.3. Ablation Study on IoU Hyperparameter λ

To investigate the impact of the IoU loss weighting coefficient λ on model performance, we conducted a series of ablation experiments across the LUNA16, Node21, and LungCT datasets. The results, as presented in Table 5, show that the value of λ significantly affects detection accuracy and localization quality. To further understand the impact of λ on model performance, we performed a sensitivity analysis and provided guidelines for selecting λ in different clinical settings.

#### 4.5.4. Sensitivity Analysis

We conducted a series of experiments where λ was varied across a range to analyze its impact on model performance. In these experiments, we tested values of λ within the range of [0.001, 1] to evaluate key performance indicators.

•Too small: a λ (e.g., 0.001) underweights the regression loss, degrading both classification and localization performance.•Too large: a λ (e.g., 0.5 or 1) overemphasizes bounding box regression, which reduces classification quality and overall robustness.•λ=0.1 outperforms all other settings across every metric and dataset, indicating that it provides the most favorable trade-off.

#### 4.5.5. Clinical Application Guidelines for Selecting λ

Based on the consistent superiority of λ=0.1 across all datasets and evaluation metrics, we recommend using this value as the default setting for various clinical diagnostic tasks. For early disease screening, where maximizing recall is crucial to minimize the risk of missed detections, λ=0.1 offers optimal sensitivity. In tasks requiring precise localization, such as lesion boundary delineation or pre-surgical planning, λ=0.1 also achieves the best bounding box regression accuracy. Moreover, for general medical image analysis tasks that demand a balanced performance between precision and recall, λ=0.1 consistently delivers robust and reliable results. Therefore, we suggest λ=0.1 as a universal setting across diverse medical imaging scenarios.

### 4.6. Comparison of Attention Mechanisms

To evaluate the effect of different attention mechanisms on detection performance, we conducted a comparative study across three datasets (LUNA16, Node21, and LungCT), integrating five representative modules: SE [25], CA [3], SIMAM [4], MCA [5], and our proposed spatial squeeze-and-excitation (SSE). The results are summarized in Table 6.

As illustrated in Table 6, the proposed spatial squeeze-and-excitation (SSE) module consistently achieves superior or competitive performance across all datasets and evaluation metrics. On the LUNA16 dataset, SSE outperforms all baselines in precision (0.781), recall (77.3%), and mAP@50 (75.5%), while also yielding the highest mAP@50–90 (33.3%). This reflects its effectiveness in enhancing localization accuracy for small nodules by focusing on spatially relevant features. A paired t-test on mAP@50–90 across five independent runs shows that SSE significantly outperforms the second-best attention (CA).

On the Node21 and LungCT datasets, SSE again surpasses other attention mechanisms, achieving the best results in all four metrics. For example, on Node21, it reaches precision of 0.895 and a mAP@50–90 of 51.4%, outperforming the next-best CA module by 2.4% and 1.2%, respectively. The observed gains are statistically significant for mAP@50–90. Similarly, on LungCT, SSE improves mAP@50–90 to 53.8%, showing strong generalization to more complex and heterogeneous nodular structures.

In contrast, modules such as SIMAM and MCA consistently lag, particularly in precision and mAP@50, indicating suboptimal spatial discrimination in the context of dense or small-object environments. While SE performs reasonably well and offers high recall, it lacks the spatial selectivity necessary for precise localization, especially under cluttered backgrounds.

These findings demonstrate that combining spatial and channel attention, as carried out in SSE, enables the model to emphasize critical lesion regions better while suppressing irrelevant context. The lightweight nature of SSE, along with its consistent improvements across diverse datasets, underscores its suitability for medical object detection tasks where fine-grained spatial awareness is essential.

### 4.7. Comparison with Mainstream Detectors

To comprehensively validate the performance of the proposed YOLOV11-EAR framework, we conducted quantitative comparisons against two widely used object detectors: faster R-CNN [26] and RetinaNet [27]. These experiments were performed on three benchmark datasets for pulmonary nodule detection, LUNA16, Node21, and LungCT, covering a variety of imaging conditions and nodule morphologies. The evaluation metrics include precision, recall, and mean average precision at IoU threshold 0.5 (mAP@50).

As shown in Table 7, YOLOV11-EAR consistently outperforms all baselines across all three datasets in terms of precision, recall, and mAP@50. On the LUNA16 dataset, our model achieves mAP of 75.5%, representing a 5.4% gain over faster R-CNN and a 6.6% gain over RetinaNet. Similar improvements are observed on Node21 and LungCT datasets, confirming the robustness and superior generalization capability of our framework.

These improvements can be attributed to the synergistic design of the spatial squeeze-and-excitation (SSE) module and the explicit aspect ratio penalty IoU (EAPIoU) loss function. The former enhances the spatial–semantic representation of nodules, while the latter enforces stronger geometric constraints, particularly benefiting the localization of irregularly shaped pulmonary nodules.

The superior performance across benchmarks validates YOLOV11-EAR as a robust and generalizable solution for real-world clinical applications, especially for detecting small and irregular pulmonary nodules under diverse acquisition settings.

### 4.8. Kernel Size Sensitivity Analysis in the SSE Module

To evaluate the impact of the convolutional kernel size in the spatial squeeze-and-excitation (SSE) module, we conducted a sensitivity analysis on the LUNA16 dataset. Specifically, we varied the kernel size of the spatial attention sub-module among {3, 5, 7, 9, and 11} to examine how different receptive field sizes affect detection performance.

During the experiments, all other network structures and training hyperparameters were kept consistent. Only the kernel size in the spatial attention branch was changed to ensure a fair comparison. Table 8 reports the model performance in terms of precision, recall, mAP@50, and mAP@50–90 under different kernel sizes.

As shown in Table 8, the model achieves the best overall performance when the kernel size is set to seven, attaining the highest values across all four metrics. This indicates that kernel size plays a critical role in the spatial attention mechanism. While smaller kernels may be inadequate for capturing broader spatial dependencies, huge kernels can introduce noise and diminish generalization. Therefore, selecting an appropriate kernel size is crucial for maximizing spatial representation capacity and enhancing detection accuracy in medical image analysis.

### 4.9. Generalizability Across Detection Frameworks

To rigorously evaluate the generalizability of the proposed spatial squeeze-and-excitation (SSE) module and explicit aspect ratio penalty IoU (EAPIoU) loss, we conducted transfer experiments by integrating them into two representative object detection frameworks: faster R-CNN and RetinaNet. These models were selected due to their structural diversity and widespread adoption in medical object detection tasks, thereby providing a robust testbed for assessing architectural independence.

All models were trained and evaluated on the LUNA16 dataset under identical training protocols with a shared ResNet-50 backbone to ensure fairness. Each framework was tested under four configurations: (i) the original baseline, (ii) integrated with SSE, (iii) using the EAPIoU loss, and (iv) combining both SSE and EAPIoU. Evaluation metrics included precision, recall, and mAP@50, consistent with prior benchmarks.

As shown in Table 9, integrating the SSE module and EAPIoU loss into both faster R-CNN and RetinaNet yields consistent performance gains across all evaluation metrics. For instance, in the faster R-CNN framework, introducing SSE alone improves mAP@50 from 70.1% to 73.1%, and EAPIoU alone raises it to 72.0%. The combined use of both modules achieves the best result of 74.7%, indicating a cumulative effect. A similar trend is observed with RetinaNet, where the complete configuration also yields a notable increase over the baseline.

These results demonstrate that the SSE module and EAPIoU loss are not tied to the YOLOV11-EAR architecture but can generalize well to different detection backbones and heads. The consistent improvements further support their modularity, robustness, and potential for broader adoption in various medical object detection tasks.

## 5. Discussion

This study addresses the challenge of detecting small and morphologically diverse pulmonary nodules in CT images by proposing the YOLOV11-EAR framework, which integrates a spatial squeeze-and-excitation (SSE) attention module and a novel explicit aspect ratio penalty IoU (EAPIoU) loss. We evaluate the model comprehensively on three publicly available datasets, LUNA16, Node21, and LungCT, covering various imaging conditions and nodule types. Experimental results demonstrate that the proposed method consistently outperforms existing mainstream methods in terms of precision, recall, and mAP, confirming its robustness and generalizability.

Our results are consistent with recent studies demonstrating the effectiveness of attention mechanisms in medical image feature extraction. Previous works such as SE-ResNeXt-50 [2] and CSE-GAN [1] mainly rely on channel attention and report improved detection performance. However, our findings suggest that channel attention alone is insufficient for retaining fine-grained spatial features. The SSE module addresses this issue by introducing spatial context, enabling more accurate localization of small and irregular nodules across diverse datasets.

Although the SSE module bears conceptual similarity to dual-attention mechanisms such as CBAM [28] and BAM [29], it differs significantly in three aspects: (1) SSE is specifically tailored for medical imaging tasks, emphasizing boundary preservation and local structure sensitivity; (2) unlike CBAM, which applies channel and spatial attention sequentially, SSE performs spatial squeezing and excitation in parallel and fuses it with channel-wise recalibration through additive interactions, achieving more stable optimization in low-contrast scenarios; (3) SSE adopts a lightweight design, ensuring minimal computational overhead when integrated into real-time detection frameworks.

To further evaluate the practical performance of our method, we conducted comparative experiments with two representative detection frameworks: faster R-CNN and RetinaNet. As shown in Table 7, YOLOV11-EAR consistently outperforms both models across all metrics and datasets. For instance, on the LungCT dataset, YOLOV11-EAR achieves mAP@50 of 0.924, significantly surpassing RetinaNet’s 0.882. This highlights the superior efficiency of our framework and the advantage of SSE and EAPIoU in capturing detailed structures and precise boundary fitting. These results demonstrate that YOLOV11-EAR not only performs well in lightweight scenarios but also surpasses mainstream models in general detection tasks.

In terms of regression loss, various IoU-based loss functions have been proposed in recent years, including GIoU [8], CIoU [9], and DIoU [10], which have significantly improved localization accuracy. GIoU introduces the smallest enclosing box to address gradient issues when boxes do not overlap but struggle with aspect ratio variation. CIoU enhances GIoU by incorporating center distance and aspect ratio penalties yet remains insufficient for shape-inconsistent targets. DIoU focuses on center distance but still relies on angular penalties for aspect ratio regularization. These approaches fail to resolve shape consistency issues, especially for non-rectangular objects, adequately. To this end, we propose EAPIoU, which introduces a squared aspect ratio penalty that directly measures the discrepancy between predicted and ground-truth boxes. EAPIoU offers stronger gradient feedback, improved shape optimization, and scale invariance, demonstrating superior generalization compared to CIoU and DIoU.

Interestingly, ablation experiments reveal that although SIoU achieves the highest recall, its mAP under stricter IoU thresholds is still inferior to EAPIoU. This suggests that SIoU’s emphasis on optimization dynamics may come at the expense of geometric precision, which is critical in medical imaging tasks that require accurate boundary delineation.

To verify the architecture-agnostic property of the proposed modules, we also integrated SSE and EAPIoU into faster R-CNN and RetinaNet for extended experiments. Results show consistent performance improvements across both frameworks, confirming the modules’ transferability and adaptability in diverse detection architectures. This further validates that SSE and EAPIoU are generic components rather than architecture-specific designs.

This study highlights the significance of shape-sensitive loss functions and spatially aware attention mechanisms in improving detection for low-contrast and small-object scenarios. The synergistic design of SSE and EAPIoU provides a comprehensive improvement in both feature localization and bounding box regression. Their robustness across multiple datasets and detection frameworks highlights their broad applicability in real-world medical settings.

Theoretically, our work challenges prevailing assumptions in object detection design, particularly the notion that aspect ratio mismatches can be effectively addressed via angular penalties. Our findings suggest that explicit ratio modeling is more effective in small-object detection tasks and should be emphasized in future loss function designs.

Despite its strengths, our method has several limitations. First, the datasets used (LUNA16, Node21, and LungCT) primarily include nodules ≥3 mm in diameter. Further validation is required on datasets with extremely small or heterogeneous lesions. Second, the training pipeline is fully supervised, limiting its applicability in scenarios with scarce annotations or significant domain shifts.

Finally, although the model remains lightweight, further optimization is required for real-time deployment on mobile or edge devices. In future work, we plan to explore semi-supervised learning and domain adaptation techniques to enhance transferability and conduct broader evaluations on datasets such as LIDC-IDRI [30] and private clinical repositories. We also aim to integrate more efficient attention mechanisms to reduce computational cost while maintaining high accuracy.

## 6. Conclusions

In this work, we proposed a YOLOV11-based method for pulmonary nodule detection, targeting improved accuracy for nodules. A novel attention mechanism combining spatial and channel attention was introduced to enhance focus on critical local regions, effectively addressing the limitations of traditional SE-based modules. Experiments showed that our module outperformed SE, coordinate attention, SIMAM, and multi-head channel attention in terms of precision and mAP@50.

We also introduced the explicit aspect ratio penalty IoU (EAPIoU) loss, which explicitly penalizes aspect ratio differences between predicted and ground-truth boxes. This design significantly improved localization accuracy, particularly for small targets, and ablation studies confirmed its superiority over other IoU-based losses. Hyperparameter analysis further validated the stability and robustness of the proposed loss.

Despite strong performance, our method’s generalization to diverse datasets and real-time deployment remains to be explored. Future work will focus on integrating semi- or self-supervised learning to utilize unlabeled data and developing lighter attention modules to reduce computational costs for broader clinical use.

## Figures and Tables

**Figure 1 sensors-25-04245-f001:**
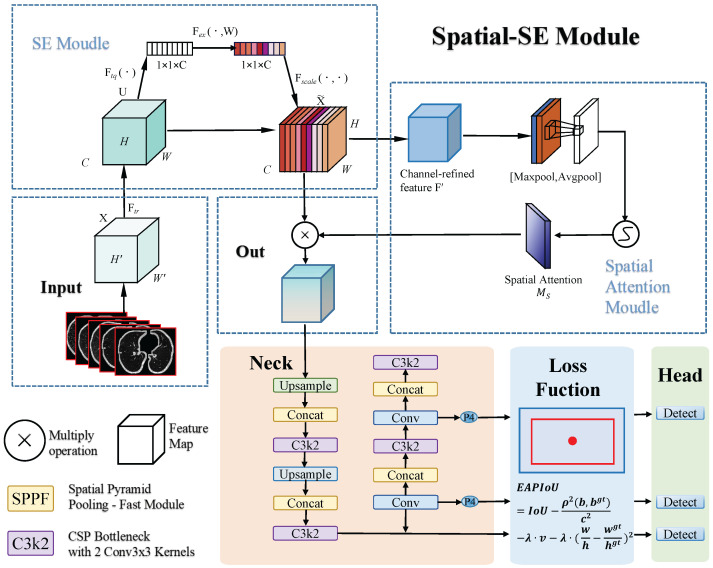
This paper presents a pulmonary nodule detection framework based on YOLOv11. First, the input lung CT slice images are processed by the SE module to adjust feature map channel weights via channel attention. The feature maps then pass through the spatial-SE module, which applies max and average pooling to extract spatial information, generating a spatial attention map that is fused with the channel-attention-processed feature maps to improve the detection of small nodules. Next, the neck module performs feature fusion and upsampling, providing multi-scale features for detection. Finally, the EAPIoU loss function applies a squared penalty to the aspect ratio differences, optimizing bounding box regression and improving the detection accuracy of small nodules. The detection head outputs the location and class of pulmonary nodules using these refined features.

**Figure 2 sensors-25-04245-f002:**
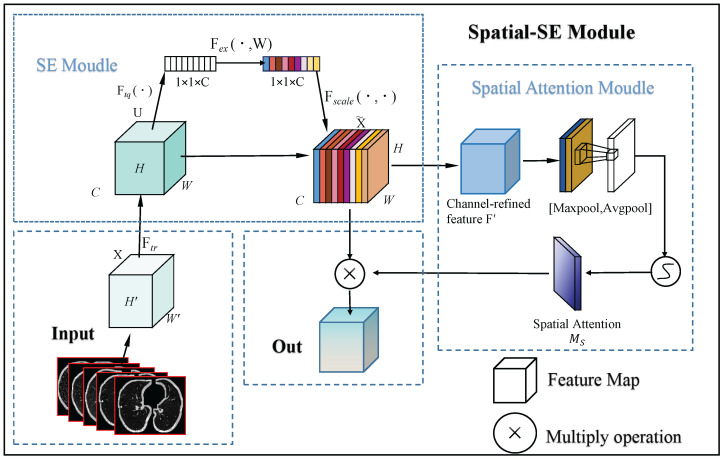
This flowchart illustrates the process of the spatial SE Module. Initially, the input feature map passes through the SE module, where global average pooling and fully connected layers calculate attention weights for each channel, refining the feature map by channel-wise scaling. Next, the SSE module utilizes max and average pooling to extract spatial data, creating a spatial attention map that emphasizes crucial spatial areas. The final output is the feature map, enhanced by both channel and spatial attention, which improves small object detection.

**Figure 3 sensors-25-04245-f003:**
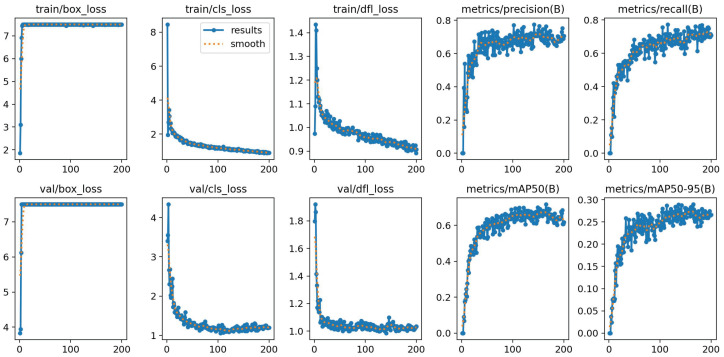
Training and validation curves of YOLOV11-EAR across 200 epochs on the LUNA16 dataset. The plots include box loss, classification loss, distribution focal loss, as well as precision, recall, and mAP metrics.

**Figure 4 sensors-25-04245-f004:**
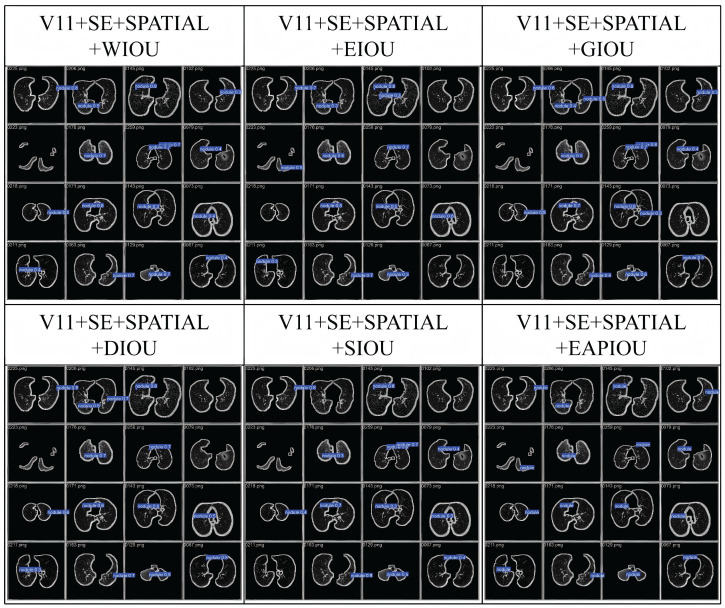
Comparison of the detection process using IoU functions in LUNA16.

**Figure 5 sensors-25-04245-f005:**
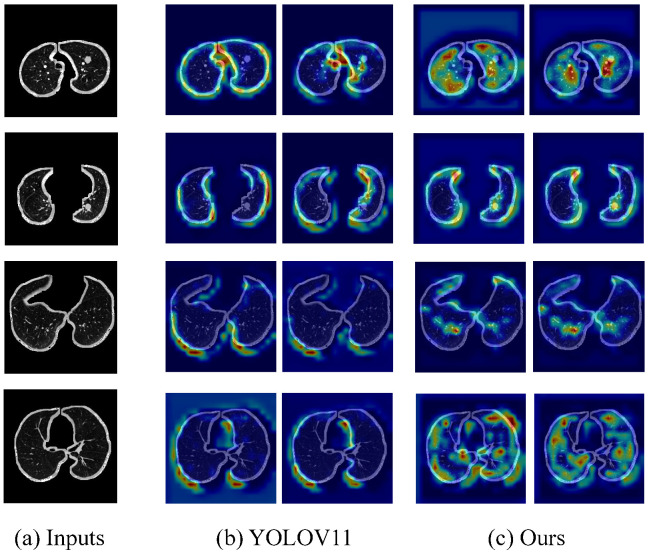
Comparison of heat maps between YOLOv11 and YOLOV11+SSE+EAPIOU.

**Table 1 sensors-25-04245-t001:** Experimental environment and training configuration.

Component	Specification	Hyperparameter	Value
Operating System	Ubuntu 20.04	Input Size	640×640
CPU	Intel Xeon W-2255 @ 3.70 GHz	Batch Size	8
GPU	2 × NVIDIA RTX A5000 (24 GB)	Epochs	200
RAM	128GB DDR4	Optimizer	SGD
CUDA Version	11.1	Learning Rate	YOLOV11 default
Framework	PyTorch 1.8.0	Data Augmentation	Flip, Scale, Normalize
		Data Loader Workers	4

**Table 2 sensors-25-04245-t002:** Comparison experiments across three datasets: LUNA16 [11], Node21 [12], and LungCT [6], with 95% confidence intervals over 5 runs.

Dataset	Model	Parameters	GFLOPs	Precision (%)	Recall (%)	mAP@50 (%)	mAP@50–90 (%)
LUNA16	YOLOV3-tiny [13]	12.1M	18.9	71.9 ± 0.10	58.0 ± 0.09	61.2 ± 0.11	26.9 ± 0.10
	YOLOV4 [14]	52.5M	119.7	63.6 ± 0.11	54.6 ± 0.10	57.9 ± 0.12	21.8 ± 0.10
	YOLOV5 [15]	2.5M	7.1	70.1 ± 0.09	68.1 ± 0.08	68.5 ± 0.09	28.8 ± 0.08
	YOLOV6 [16]	4.2M	11.8	74.7 ± 0.08	65.6 ± 0.09	67.1 ± 0.08	27.7 ± 0.09
	YOLOV7 [17]	37.2M	105.1	61.3 ± 0.10	61.2 ± 0.09	58.7 ± 0.10	23.4 ± 0.09
	YOLOV8 [18]	3.0M	8.1	71.3 ± 0.07	73.0 ± 0.08	72.8 ± 0.07	30.8 ± 0.08
	YOLOV9 [19]	2.0M	7.6	75.4 ± 0.08	67.0 ± 0.07	65.8 ± 0.08	29.3 ± 0.07
	YOLOV10 [20]	20.4M	97.9	75.6 ± 0.09	64.9 ± 0.08	70.1 ± 0.07	30.2 ± 0.09
	YOLOV11 [21]	2.5M	6.3	69.1 ± 0.07	70.6 ± 0.07	66.1 ± 0.08	29.2 ± 0.07
	Ours	2.5M	6.3	**78.1 ± 0.05**	**77.3 ± 0.07**	**75.5 ± 0.06**	**33.3 ± 0.05**
Node21	YOLOV3-tiny [13]	12.1M	18.9	86.5 ± 0.07	77.8 ± 0.08	84.1 ± 0.06	46.2 ± 0.07
	YOLOV4 [14]	52.5M	119.7	84.2 ± 0.09	75.1 ± 0.07	82.5 ± 0.08	44.0 ± 0.07
	YOLOV5 [15]	2.5M	7.1	86.8 ± 0.07	78.2 ± 0.05	85.6 ± 0.06	47.5 ± 0.07
	YOLOV6 [16]	4.2M	11.8	87.3 ± 0.06	79.0 ± 0.07	86.2 ± 0.05	48.1 ± 0.08
	YOLOV7 [17]	37.2M	105.1	85.9 ± 0.04	76.4 ± 0.06	83.8 ± 0.05	46.5 ± 0.08
	YOLOV8 [18]	3.0M	8.1	88.2 ± 0.06	80.1 ± 0.08	87.5 ± 0.07	49.3 ± 0.05
	YOLOV9 [19]	2.0M	7.6	88.4 ± 0.07	80.8 ± 0.06	87.6 ± 0.04	49.6 ± 0.08
	YOLOV10 [20]	20.4M	97.9	87.7 ± 0.06	79.6 ± 0.09	86.9 ± 0.09	48.4 ± 0.07
	YOLOV11 [21]	2.5M	6.3	87.9 ± 0.08	79.9 ± 0.06	87.2 ± 0.04	49.0 ± 0.08
	Ours	2.5M	6.3	**89.5 ± 0.07**	**82.5 ± 0.05**	**89.1 ± 0.05**	**51.4 ± 0.06**
LungCT	YOLOV3-tiny [13]	12.1M	18.9	87.7 ± 0.05	79.5 ± 0.08	87.1 ± 0.07	49.3 ± 0.08
	YOLOV4 [14]	52.5M	119.7	86.4 ± 0.06	78.3 ± 0.04	85.2 ± 0.08	47.0 ± 0.09
	YOLOV5 [15]	2.5M	7.1	87.4 ± 0.05	79.5 ± 0.07	89.2 ± 0.07	52.1 ± 0.08
	YOLOV6 [16]	4.2M	11.8	87.0 ± 0.07	80.0 ± 0.09	88.6 ± 0.06	50.6 ± 0.07
	YOLOV7 [17]	37.2M	105.1	86.1 ± 0.08	78.5 ± 0.08	86.7 ± 0.06	48.3 ± 0.09
	YOLOV8 [18]	3.0M	8.1	88.0 ± 0.09	81.6 ± 0.05	90.5 ± 0.04	53.3 ± 0.08
	YOLOV9 [19]	2.0M	7.6	88.1 ± 0.06	81.0 ± 0.09	90.2 ± 0.07	52.8 ± 0.08
	YOLOV10 [20]	20.4M	97.9	87.6 ± 0.09	80.3 ± 0.06	89.7 ± 0.07	51.9 ± 0.06
	YOLOV11 [21]	2.5M	6.3	87.8 ± 0.06	80.6 ± 0.08	90.1 ± 0.09	52.4 ± 0.07
	Ours	2.5M	6.3	**88.7 ± 0.04**	**82.5 ± 0.06**	**92.4 ± 0.08**	**53.8 ± 0.09**

**Table 3 sensors-25-04245-t003:** Ablation experiment of different IoU loss variants across datasets with 95% confidence intervals.

Dataset	IoU Variant	Precision (%)	Recall (%)	mAP@50 (%)	mAP@50–90 (%)
LUNA16	WIoU [22]	74.7 ± 0.05	73.1 ± 0.04	72.0 ± 0.06	30.4 ± 0.04
	EIoU [23]	72.7 ± 0.04	72.3 ± 0.05	70.9 ± 0.04	30.2 ± 0.05
	GIoU [8]	73.3 ± 0.05	71.4 ± 0.03	71.5 ± 0.05	30.3 ± 0.04
	DIoU [10]	77.3 ± 0.03	74.8 ± 0.04	73.5 ± 0.05	30.4 ± 0.03
	SIoU [24]	76.5 ± 0.06	**77.3 ± 0.04**	74.0 ± 0.05	31.1 ± 0.02
	EAPIoU	**78.1 ± 0.05**	**77.3 ± 0.07**	**75.5 ± 0.06**	**33.3 ± 0.05**
Node21	WIoU [22]	86.1 ± 0.04	78.2 ± 0.05	85.6 ± 0.07	48.1 ± 0.05
	EIoU [23]	85.3 ± 0.05	77.0 ± 0.03	84.1 ± 0.02	47.5 ± 0.05
	GIoU [8]	84.9 ± 0.06	76.5 ± 0.04	83.7 ± 0.03	47.1 ± 0.02
	DIoU [10]	86.5 ± 0.04	79.1 ± 0.04	86.5 ± 0.05	48.4 ± 0.03
	SIoU [24]	87.2 ± 0.03	80.1 ± 0.06	87.2 ± 0.04	49.0 ± 0.04
	EAPIoU	**89.5 ± 0.07**	**82.5 ± 0.05**	**89.1 ± 0.05**	**51.4 ± 0.06**
LungCT	WIoU [22]	86.4 ± 0.04	79.2 ± 0.04	88.1 ± 0.03	50.1 ± 0.05
	EIoU [23]	85.7 ± 0.02	78.0 ± 0.03	87.0 ± 0.05	49.5 ± 0.04
	GIoU [8]	85.9 ± 0.04	77.8 ± 0.06	87.2 ± 0.06	49.8 ± 0.06
	DIoU [10]	87.3 ± 0.05	80.4 ± 0.04	89.0 ± 0.06	51.2 ± 0.02
	SIoU [24]	87.5 ± 0.06	80.9 ± 0.02	89.5 ± 0.03	51.9 ± 0.03
	EAPIoU	**88.7 ± 0.04**	**82.5 ± 0.06**	**92.4 ± 0.08**	**53.8 ± 0.09**

**Table 4 sensors-25-04245-t004:** Comparison of module ablation experiment results with diverse confidence intervals.

Datasets	SSE	EAPIoU	Precision (%)	Recall (%)	mAP@50 (%)	mAP@50–90 (%)
LUNA16	–	–	67.2 ± 0.06	68.5 ± 0.08	65.2 ± 0.05	28.7 ± 0.10
	✓	–	72.5 ± 0.05	71.0 ± 0.06	70.1 ± 0.04	30.1 ± 0.07
	–	✓	70.9 ± 0.07	70.2 ± 0.09	68.9 ± 0.05	29.5 ± 0.06
	✓	✓	**78.1 ± 0.05**	**77.3 ± 0.07**	**75.5 ± 0.06**	**33.3 ± 0.05**
Node21	–	–	86.1 ± 0.05	78.0 ± 0.07	85.2 ± 0.06	48.6 ± 0.08
	✓	–	87.7 ± 0.04	79.5 ± 0.06	87.0 ± 0.05	49.3 ± 0.06
	–	✓	88.4 ± 0.05	81.1 ± 0.07	88.4 ± 0.04	50.0 ± 0.05
	✓	✓	**89.5 ± 0.07**	**82.5 ± 0.05**	**89.1 ± 0.05**	**51.4 ± 0.06**
LungCT	–	–	83.2 ± 0.07	80.1 ± 0.09	85.3 ± 0.05	48.5 ± 0.06
	✓	–	85.8 ± 0.05	81.6 ± 0.06	88.2 ± 0.04	49.6 ± 0.06
	–	✓	87.3 ± 0.04	81.9 ± 0.05	90.1 ± 0.03	51.2 ± 0.04
	✓	✓	**88.7 ± 0.04**	**82.5 ± 0.06**	**92.4 ± 0.08**	**53.8 ± 0.09**

**Table 5 sensors-25-04245-t005:** IoU hyperparameter λ experiment across datasets with 95% confidence intervals.

Dataset	λ	Precision (%)	Recall (%)	mAP@50 (%)	mAP@50–90 (%)
LUNA16	0.001	67.5 ± 0.11	68.0 ± 0.10	66.1 ± 0.09	28.9 ± 0.11
	0.01	70.1 ± 0.09	71.2 ± 0.09	69.8 ± 0.08	30.5 ± 0.08
	0.1	**78.1 ± 0.05**	**77.3 ± 0.07**	**75.5 ± 0.06**	**33.3 ± 0.05**
	0.5	70.7 ± 0.10	69.5 ± 0.11	70.3 ± 0.09	27.8 ± 0.10
	1	68.9 ± 0.11	72.9 ± 0.10	70.5 ± 0.09	27.7 ± 0.11
Node21	0.001	69.0 ± 0.10	70.2 ± 0.09	67.9 ± 0.06	30.3 ± 0.07
	0.01	71.9 ± 0.09	73.0 ± 0.08	70.7 ± 0.06	31.2 ± 0.07
	0.1	**89.5 ± 0.07**	**82.5 ± 0.05**	**89.1 ± 0.05**	**51.4 ± 0.06**
	0.5	73.3 ± 0.08	74.5 ± 0.05	72.9 ± 0.07	47.2 ± 0.09
	1	70.5 ± 0.07	74.1 ± 0.09	71.9 ± 0.08	28.8 ± 0.06
LungCT	0.001	71.1 ± 0.10	71.3 ± 0.09	69.2 ± 0.08	30.9 ± 0.06
	0.01	74.2 ± 0.07	74.1 ± 0.08	72.9 ± 0.05	32.1 ± 0.06
	0.1	**88.7 ± 0.04**	**82.5 ± 0.06**	**92.4 ± 0.08**	**53.8 ± 0.09**
	0.5	76.3 ± 0.07	77.0 ± 0.08	84.6 ± 0.06	50.4 ± 0.07
	1	72.6 ± 0.10	75.5 ± 0.06	74.0 ± 0.07	29.6 ± 0.08

**Table 6 sensors-25-04245-t006:** Variational attention mechanism experiments across datasets with 95% confidence intervals.

Dataset	Attention	Precision (%)	Recall (%)	mAP@50 (%)	mAP@50–90 (%)
LUNA16	SE [25]	74.1 ± 0.04	72.1 ± 0.05	71.0 ± 0.06	30.5 ± 0.04
	CA [3]	74.3 ± 0.03	71.8 ± 0.06	71.4 ± 0.05	**31.0 ± 0.03**
	SIMAM [4]	71.9 ± 0.06	69.5 ± 0.06	69.2 ± 0.04	29.8 ± 0.05
	MCA [5]	73.2 ± 0.04	70.1 ± 0.07	70.5 ± 0.05	28.4 ± 0.02
	SSE	**78.1 ± 0.05**	**77.3 ± 0.07**	**75.5 ± 0.06**	**33.3 ± 0.05**
Node21	SE [25]	86.8 ± 0.05	79.9 ± 0.04	86.4 ± 0.06	49.8 ± 0.05
	CA [3]	87.1 ± 0.03	80.2 ± 0.06	86.9 ± 0.06	50.2 ± 0.03
	SIMAM [4]	85.1 ± 0.04	77.5 ± 0.07	85.2 ± 0.06	48.0 ± 0.05
	MCA [5]	85.9 ± 0.04	78.6 ± 0.07	85.8 ± 0.06	47.5 ± 0.05
	SSE	**89.5 ± 0.07**	**82.5 ± 0.05**	**89.1 ± 0.05**	**51.4 ± 0.06**
LungCT	SE [25]	86.4 ± 0.04	80.8 ± 0.06	90.2 ± 0.03	52.0 ± 0.07
	CA [3]	86.6 ± 0.06	81.1 ± 0.03	90.5 ± 0.06	52.6 ± 0.03
	SIMAM [4]	84.3 ± 0.07	78.2 ± 0.02	88.6 ± 0.04	50.1 ± 0.04
	MCA [5]	84.9 ± 0.04	78.9 ± 0.04	89.0 ± 0.02	49.8 ± 0.05
	SSE	**88.7 ± 0.04**	**82.5 ± 0.06**	**92.4 ± 0.08**	**53.8 ± 0.09**

**Table 7 sensors-25-04245-t007:** Comparative results on three public datasets with 95% confidence intervals (best results are marked in **bold**).

Dataset	Method	Precision (%)	Recall (%)	mAP@50 (%)
LUNA16	RetinaNet	70.9 ± 0.07	70.2 ± 0.10	68.9 ± 0.09
	Faster R-CNN	72.5 ± 0.10	71.0 ± 0.09	70.1 ± 0.06
	YOLOV11-EAR	**78.1 ± 0.05**	**77.3 ± 0.07**	**75.5 ± 0.06**
Node21	RetinaNet	87.7 ± 0.09	79.5 ± 0.08	87.0 ± 0.10
	Faster R-CNN	88.4 ± 0.06	81.1 ± 0.10	88.4 ± 0.07
	YOLOV11-EAR	**89.5 ± 0.07**	**82.5 ± 0.05**	**89.1 ± 0.05**
LungCT	RetinaNet	85.8 ± 0.10	81.6 ± 0.07	88.2 ± 0.09
	Faster R-CNN	87.3 ± 0.09	81.9 ± 0.11	90.1 ± 0.08
	YOLOV11-EAR	**88.7 ± 0.04**	**82.5 ± 0.06**	**92.4 ± 0.08**

**Table 8 sensors-25-04245-t008:** Detection performance under different convolutional kernel sizes in the SSE module (LUNA16 dataset) with 95% confidence intervals.

Kernel Size	Precision (%)	Recall (%)	mAP@50 (%)	mAP@50–90 (%)
3	75.4 ± 0.06	71.9 ± 0.11	69.6 ± 0.07	30.1 ± 0.08
5	76.7 ± 0.05	75.2 ± 0.07	72.1 ± 0.06	31.7 ± 0.07
7	**78.1 ± 0.05**	**77.3 ± 0.07**	**75.5 ± 0.06**	**33.3 ± 0.05**
9	77.2 ± 0.11	76.0 ± 0.08	73.9 ± 0.06	32.8 ± 0.07
11	76.3 ± 0.10	75.5 ± 0.09	72.6 ± 0.10	31.9 ± 0.08

**Table 9 sensors-25-04245-t009:** Ablation study of SSE and EAPIoU across different detection frameworks (LUNA16 Dataset) with 95% confidence intervals.

Model	SSE	EAPIoU	Precision (%)	Recall (%)	mAP@50 (%)
Faster R-CNN	–	–	72.5 ± 0.10	71.0 ± 0.09	70.1 ± 0.06
	✓	–	75.2 ± 0.08	73.5 ± 0.08	73.1 ± 0.09
	–	✓	74.1 ± 0.09	72.6 ± 0.08	72.0 ± 0.09
	✓	✓	**76.9 ± 0.07**	**76.0 ± 0.07**	**74.7 ± 0.08**
RetinaNet	–	–	70.9 ± 0.07	70.2 ± 0.10	68.9 ± 0.09
	✓	–	73.5 ± 0.07	72.4 ± 0.06	71.0 ± 0.05
	–	✓	72.5 ± 0.08	71.6 ± 0.07	70.3 ± 0.08
	✓	✓	**74.1 ± 0.05**	**73.8 ± 0.06**	**72.2 ± 0.06**

## Data Availability

The original contributions presented in this study are included in the article. Further inquiries can be directed to the corresponding author.

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
