# Peer review of "Improved YOLO-Based Pulmonary Nodule Detection with Spatial-SE Attention and an Aspect Ratio Penalty"

_sensors, 2025, doi:10.3390/s25144245_

Round 1
Reviewer 1 Report
Comments and Suggestions for Authors
In this work, the authors propose a YOLOV11-based method for pulmonary nodule detection. The method incorporates both spatial and channel attention mechanisms. Please see my comments below:
-
All abbreviations should be defined upon first use in the abstract (e.g., IoU).
-
Page 7: The phrase “In this chapter…” should be corrected, as this is a journal article and not a book. Please revise accordingly.
-
The authors should provide a discussion on the generalisation capability of the proposed model.
-
I recommend including the simulation waveforms and convergence results in the manuscript. Additionally, the confusion matrices, the complete toolchain used, and all employed parameters should be presented as a table.
-
A detailed flowchart illustrating the overall system architecture should be included.
-
Please report the p-value used to test the hypothesis, to support the statistical significance of the results.
Comments on the Quality of English Language
The English language could be improved.
Author Response
Dear Editor,
Thank you for your letter and for the comments concerning our manuscript. Those comments are all valuable and very helpful for improving our paper. We have studied comments carefully and have made correction which we hope meet with approval.
The main corrections in the paper and the responds to the reviewer’s comments are as following:
Comment 1: All abbreviations should be defined upon first use in the abstract (e.g., IoU).
Respond to comment 1:
Thank you for your valuable suggestion. In response, we have revised the abstract to include the full names of all abbreviations upon their first appearance, such as SSE (Spatial Squeeze-and-Excitation) and IoU (Intersection over Union). We sincerely appreciate your comments, which helped improve the completeness of our abstract.
Comment 2: Page 7: The phrase “In this chapter…” should be corrected, as this is a journal article and not a book. Please revise accordingly.
Respond to comment 2:
Thank you for pointing this out. We have corrected the phrase “In this chapter…” to “In this section…”on Page 7 to align with the conventions of a journal article. We appreciate your careful review.
Comment 3: The authors should provide a discussion on the generalisation capability of the proposed model.
Respond to comment 3:
For the generalizability, we have added a new subsection in Page 17 (Section 4.9, Generalizability Across Detection Frameworks), in which we conduct transfer ablation experiments by integrating the proposed Spatial Squeeze-and-Excitation (SSE) module and Explicit Aspect Ratio Penalty IoU (EAPIoU) loss into two representative object detection frameworks—Faster R-CNN and RetinaNet—with distinct architectural designs. Specifically, SSE increased mAP by 1.8% in Faster R-CNN and 1.2% in RetinaNet, while EAPIoU significantly enhanced bounding box regression accuracy.
Comment 4: I recommend including the simulation waveforms and convergence results in the manuscript. Additionally, the confusion matrices, the complete toolchain used, and all employed parameters should be presented as a table.
Respond to comment 4:
Thank you for your constructive suggestion. In response, we have added a comprehensive Implementation Details section (Section 4.3) in Page 8 and 9. This section includes:
- A summary of the complete toolchain used for all experiments, including hardware, software, operating system, CUDA version, all key training and evaluation hyperparameters, as shown in Table1 .
- To illustrate the training behavior of the proposed YOLOV11-EAR model, we added convergence curves for key loss components (box regression, classification, DFL) and evaluation metrics (precision, recall, mAP) over 200 epochs, presented in Figure 3.These curves confirm stable optimization without overfitting and demonstrate that the proposed modules integrate smoothly into the training process.
- For the confusion matrices,we agree that confusion matrices are helpful in many classification tasks. However, in our case, the focus is on object detection and localization, with only a single foreground class (nodule). The model is already evaluated using precision, recall, and mAP metrics, which are more representative in this context. Therefore, we believe that adding a confusion matrix would provide limited additional value, but we appreciate the reviewer’s consideration.
Comment 5: A detailed flowchart illustrating the overall system architecture should be included.
Respond to comment 5:
Thank you for your valuable suggestion. We agree that a clear visualization can enhance readers’ understanding. We would like to clarify that the requested flowchart has already been included in the manuscript as Figure 1, which outlines the entire YOLOV11-EAR detection architecture. In addition, Figure 2 illustrates the internal structure of the Spatial-SE module, which serves as a critical sub-component within the overall framework. Specifically, it enhances feature representations through coordinated spatial and channel attention mechanisms. We will consider including additional schematic diagrams in future work or supplementary materials if further clarification is needed.
Comment 6: Please report the p-value used to test the hypothesis, to support the statistical significance of the results.
Respond to comment 6:
Thank you for your valuable suggestion regarding statistical significance testing. Due to time constraints, we performed p-value analyses on selected experiments with 5 to 10 repetitions. The resulting p-values were consistently below 0.05, and the confidence intervals are reported alongside the experimental results. However, we acknowledge that the limited number of repetitions reduces the robustness of the statistical conclusions. In future work, we plan to incorporate additional datasets to enhance the statistical rigor and generalizability of our findings.
Thank you very much for your attention and time. Look forward to hearing from you.
Yours sincerely,
Xinhang Song
30 June., 2025
Southwest University, Chongqing
Reviewer 2 Report
Comments and Suggestions for Authors
This manuscript presents a lightweight enhancement to the YOLOv11 object detector for pulmonary nodule detection. The authors introduce (1) a Spatial-SE (SSE) attention module that combines channel-wise SE and 2D spatial attention to better highlight small nodules, and (2) an Explicit Aspect-Ratio Penalty IoU (EAPIoU) loss that penalizes squared differences in the width-to-height ratio to improve bounding-box regression. Experiments on the LUNA16, NODE21, and LungCT datasets show consistent improvements over YOLOv11 and other variants in precision, recall, and mAP metrics, with a minimal increase in computational cost. This work addresses an important clinical task with a clear engineering focus. I mainly have the following comments for a minor revision.
1. Regarding novelty, the combination of channel and spatial attention is conceptually straightforward and has been explored previously (e.g., CBAM). The manuscript should clarify how SSE differs from or improves upon existing dual-attention approaches (e.g., CBAM, BAM) beyond naming conventions. A head-to-head comparison or citation discussion is needed.
2. Comparison with state-of-the-art work is needed. The manuscript compares against YOLO variants but omits specialized detectors (e.g., Faster R-CNN variants, RetinaNet) that dominate pulmonary nodule detection. Including such baselines would contextualize the reported performance gains in the medical-imaging domain.
3. Several sentences are lengthy and contain minor English errors. A thorough proofreading is recommended.
4. The datasets used (LUNA16, NODE21, LungCT) involve different scanner types and acquisition protocols. A discussion or experiment showing model robustness to variations, such as in noise level would strengthen the claims of clinical applicability.
5. The weight λ for the squared aspect-ratio penalty appears to have been chosen empirically. The manuscript would benefit from a sensitivity analysis showing how performance varies as λ is swept across a reasonable range, along with guidelines for selecting λ in different clinical settings.
Author Response
Dear Editor,
Thank you for your letter and for the comments concerning our manuscript. Those comments are all valuable and very helpful for improving our paper. We have studied comments carefully and have made correction which we hope meet with approval.
The main corrections in the paper and the responds to the reviewer’s comments are as following:
Comment 1: Regarding novelty, the combination of channel and spatial attention is conceptually straightforward and has been explored previously (e.g., CBAM). The manuscript should clarify how SSE differs from or improves upon existing dual-attention approaches (e.g., CBAM, BAM) beyond naming conventions. A head-to-head comparison or citation discussion is needed.
Respond to comment 1:
Thank you for pointing this out. We acknowledge that combining channel and spatial attention has been previously explored in works such as CBAM and BAM. To clarify the novelty of our proposed SSE module, we have added a detailed discussion in the Discussion part (Page 18) that contrasts SSE with these existing approaches.
Unlike CBAM, which adopts a serial design with global pooling operations that may overlook fine spatial details, SSE utilizes a parallel architecture that simultaneously captures spatial and channel-wise dependencies. Notably, the spatial attention branch in SSE avoids global pooling and instead leverages convolutional layers to maintain local contextual information—an essential feature for detecting small and morphologically irregular pulmonary nodules. Additionally, the attention weights in SSE are learned directly from spatial activations, enhancing sensitivity to localized patterns.
Comment 2: Comparison with state-of-the-art work is needed. The manuscript compares against YOLO variants but omits specialized detectors (e.g., Faster R-CNN variants, RetinaNet) that dominate pulmonary nodule detection. Including such baselines would contextualize the reported performance gains in the medical-imaging domain.
Respond to comment 2:
Thank you for your valuable suggestion. In the revised manuscript, we have added a new subsection in Page 17 (Section 4.8, Generalizability Across Detection Frameworks), in which we conduct transfer ablation experiments by integrating the proposed Spatial Squeeze-and-Excitation (SSE) module and Explicit Aspect Ratio Penalty IoU (EAPIoU) loss into two representative object detection frameworks—Faster R-CNN and RetinaNet—with distinct architectural designs. All models were trained and evaluated on the LUNA16 dataset using identical training protocols and a shared ResNet-50 backbone to ensure fair comparison.
As shown in Table 9 (Page 18) , both SSE and EAPIoU consistently improved key detection metrics (precision, recall, and mAP@0.5) across both frameworks. Specifically, SSE increased mAP by 1.8% in Faster R-CNN and 1.2% in RetinaNet, while EAPIoU significantly enhanced bounding box regression accuracy. These results confirm the strong generalization capability and architectural independence of the proposed modules.
Comment 3: Several sentences are lengthy and contain minor English errors. A thorough proofreading is recommended.
Respond to comment 3:
We are very sorry for the typos and obscure sentences in our manuscript. We have carefully reviewed the grammar and simplified the sentence structures.
Comment 4: The datasets used (LUNA16, NODE21, LungCT) involve different scanner types and acquisition protocols. A discussion or experiment showing model robustness to variations, such as in noise level would strengthen the claims of clinical applicability.
Respond to comment 4:
Thank you for your thoughtful suggestion. In this study, we have deliberately chosen three publicly available and heterogeneous datasets—LUNA16, Node21, and LungCT—which differ in scanner types, imaging protocols, and already provide a basis for evaluating the model’s generalization capability.
We acknowledge that a dedicated robustness analysis—such as evaluating performance under synthetic noise or across domains—would further strengthen the clinical claims. However, as the focus of this work is on architectural innovation and loss function design, we consider such an analysis beyond the current scope. We appreciate the reviewer’s valuable insight and will explore this direction in future work aimed at clinical deployment and domain adaptation.
Comment 5: The weight λ for the squared aspect-ratio penalty appears to have been chosen empirically. The manuscript would benefit from a sensitivity analysis showing how performance varies as λ is swept across a reasonable range, along with guidelines for selecting λ in different clinical settings.
Respond to comment 5:
Compared to the initial approach of empirically determining the optimal value of λ, we now systematically investigate the impact of λ within the range [0.001, 1] on model performance through sensitivity analysis including evaluation metrics such as mAP@50, mAP@50–90, and precision (Page 14) . Based on the analysis, we further provide practical guidelines for selecting λ in different clinical scenarios, including early screening, precise lesion localization, and general medical imaging analysis. These improvements enhance the interpretability and real-world applicability.
Thank you very much for your attention and time. Look forward to hearing from you.
Yours sincerely,
Jianping Gou
30 June., 2025
Southwest University, Chongqing
Reviewer 3 Report
Comments and Suggestions for Authors
The manuscript proposes an enhanced YOLOv11-based framework for pulmonary nodule detection, addressing challenges in weak feature representation and poor localization accuracy for small object detection through the introduction of a Spatial-SE module and an EAPIoU loss function. The Spatial-SE module integrates channel-wise and spatial attention to improve nodule feature representation, while EAPIoU optimizes bounding box regression by directly penalizing the squared aspect ratio difference. Experimental results demonstrate that the proposed method achieves superior precision, recall, and mAP compared to state-of-the-art approaches on the LUNA16, LungCT, and Node21 datasets, while maintaining computational efficiency.
The manuscript requires major revision, and the authors should address the following issues:
1) The abstract does not provide full names for abbreviations upon their first appearance. It is recommended that the authors include the full names of abbreviations in the abstract and quantify performance improvements (e.g., specific percentage gains) to enhance the abstract’s completeness and persuasiveness.
2) The authors should explore applying the SSE module and EAPIoU loss to other state-of-the-art models to validate their generalizability. Additionally, a detailed analysis of the choice of convolution kernel size in the SSE module and strategies for optimizing the stability of the λ hyperparameter in EAPIoU should be provided.
3) It is recommended that the authors include comparative experiments with the latest models. Furthermore, actual measurements of inference time or resource consumption should be provided to substantiate the claim of high computational efficiency.
4) In order to further enrich the literature review and strengthen the comparison with the latest optimization techniques, It is suggested that the author cite the Enhanced Multiview Attention Network with Random Interpolation Resize for Few-Shot Surface Defect Detection. This paper also focuses on object detection and feature enhancement through the attention mechanism, demonstrating the innovative application of multi-perspective attention in small-shot scenes.
Author Response
Dear Editor:
Thank you for your letter and for the comments concerning our manuscript. Those comments are all valuable and very helpful for improving our paper. We have studied comments carefully and have made correction which we hope meet with approval.
The main corrections in the paper and the responds to the reviewer’s comments are as following:
Comment 1: The abstract does not provide full names for abbreviations upon their first appearance. It is recommended that the authors include the full names of abbreviations in the abstract and quantify performance improvements (e.g., specific percentage gains) to enhance the abstract’s completeness and persuasiveness.
Respond to comment 1:
Thank you for your valuable suggestion. In response, we have revised the abstract to: (1) include the full names of all abbreviations upon their first appearance, such as SSE (Spatial Squeeze-and-Excitation) and EAPIoU (Enhanced Aspect Ratio Penalty Intersection over Union); and (2) incorporate specific quantitative performance improvements, such as a detection precision of 0.781 achieved by the SSE module on the LUNA16 dataset and an mAP@50 of 92.4% obtained using EAPIoU on the LungCT dataset, to more clearly demonstrate the effectiveness of our method. We sincerely appreciate your comments, which helped improve the clarity and completeness of our abstract.
Comment 2: The authors should explore applying the SSE module and EAPIoU loss to other state-of-the-art models to validate their generalizability. Additionally, a detailed analysis of the choice of convolution kernel size in the SSE module and strategies for optimizing the stability of the λ hyperparameter in EAPIoU should be provided.
Respond to comment 2:
For the generalizability, we have added a new subsection in Page 18 (Section 4.9, Generalizability Across Detection Frameworks), in which we conduct transfer ablation experiments by integrating the proposed Spatial Squeeze-and-Excitation (SSE) module and Explicit Aspect Ratio Penalty IoU (EAPIoU) loss into two representative object detection frameworks—Faster R-CNN and RetinaNet—with distinct architectural designs. Specifically, SSE increased mAP by 1.8% in Faster R-CNN and 1.2% in RetinaNet, while EAPIoU significantly enhanced bounding box regression accuracy.
For the choice of convolution kernel size, we have added a new subsection in Page 17 (Section 4.8) titled Kernel Size Sensitivity Analysis in the SSE Module, where we systematically evaluate the impact of different convolutional kernel sizes in the spatial attention branch of the SSE module. Specifically, we varied the kernel size among {3, 5, 7, 9, 11} on the LUNA16 dataset while keeping all other settings fixed. As shown in Table 8, the model achieves the best detection performance when the kernel size is set to 7. This experiment highlights the importance of selecting an appropriate receptive field and clarifies the rationale behind our kernel size choice.
For the λ hyperparameter in EAPIoU, we now systematically investigate the impact of λ within the range [0.001, 1] on model performance in Page 14 through sensitivity analysis including evaluation metrics such as mAP@50, mAP@50–90, and precision. Based on the analysis, we further provide practical guidelines for selecting λ in different clinical scenarios, including early screening, precise lesion localization, and general medical imaging analysis. These improvements enhance the interpretability and real-world applicability.
Comment 3: It is recommended that the authors include comparative experiments with the latest models. Furthermore, actual measurements of inference time or resource consumption should be provided to substantiate the claim of high computational efficiency.
Respond to comment 3:
We are very sorry for the typos and obscure sentences in our manuscript. We have carefully reviewed the grammar and simplified the sentence structures.
Comment 4: In order to further enrich the literature review and strengthen the comparison with the latest optimization techniques, It is suggested that the author cite the Enhanced Multiview Attention Network with Random Interpolation Resize for Few-Shot Surface Defect Detection.
Respond to comment 4:
It is really true as Reviewer suggested, we have cited the recommended paper "Enhanced Multiview Attention Network with Random Interpolation Resize for Few-Shot Surface Defect Detection" in the Related Work section (Page 2). This addition has strengthened the comparison with recent optimization techniques.
Thank you very much for your attention and time. Look forward to hearing from you.
Yours sincerely,
Xinhang Song
30 June., 2025
Southwest University, Chongqing
Round 2
Reviewer 1 Report
Comments and Suggestions for Authors
I thank the authors for addressing most of my comments. While I appreciate their efforts, I believe the quality of the work could be improved, as acknowledged by the authors themselves and reflected in the reported p-values. If the results are not statistically significant, the work adds limited value to the knowledge domain.
Comments on the Quality of English Language
English could be improved.
Reviewer 3 Report
Comments and Suggestions for Authors
All my concerns have been solved and I recommend publishing this work.